# Oxidation of cellular amino acid pools leads to cytotoxic mistranslation of the genetic code

Tammy J Bullwinkle[1†], Noah M Reynolds[1,2†], Medha Raina[2,3], Adil Moghal[2,3], Eleftheria Matsa[1], Andrei Rajkovic[2,4], Huseyin Kayadibi[5‡], Farbod Fazlollahi[5], Christopher Ryan[5§], Nathaniel Howitz[6], Kym F Faull[5], Beth A Lazazzera[6], Michael Ibba[1,2,3,4]*

[1]Department of Microbiology, Ohio State University, Columbus, United States; [2]Center for RNA Biology, Ohio State University, Columbus, United States; [3]Ohio State Biochemistry Program, Ohio State University, Columbus, United States; [4]Molecular, Cellular and Developmental Biology Program, Ohio State University, Columbus, United States; [5]Pasarow Mass Spectrometry Laboratory, Semel Institute for Neuroscience and Human Behavior, University of California, Los Angeles, United States; [6]Department of Microbiology, Immunology and Molecular Genetics, University of California, Los Angeles, Los Angeles, United States

*For correspondence: ibba.1@att.net

†These authors contributed equally to this work

Present address: ‡Medical Biochemistry Laboratory, Adana Military Hospital, Adana, Turkey; §California Department of Toxic Substances Control, Los Angeles, United States

Competing interests: The authors declare that no competing interests exist.

**Abstract** Aminoacyl-tRNA synthetases use a variety of mechanisms to ensure fidelity of the genetic code and ultimately select the correct amino acids to be used in protein synthesis. The physiological necessity of these quality control mechanisms in different environments remains unclear, as the cost vs benefit of accurate protein synthesis is difficult to predict. We show that in *Escherichia coli*, a non-coded amino acid produced through oxidative damage is a significant threat to the accuracy of protein synthesis and must be cleared by phenylalanine-tRNA synthetase in order to prevent cellular toxicity caused by mis-synthesized proteins. These findings demonstrate how stress can lead to the accumulation of non-canonical amino acids that must be excluded from the proteome in order to maintain cellular viability.

## Introduction

The faithful translation of mRNA into the corresponding protein sequence is an essential step in gene expression. The accuracy of translation depends on the precise pairing of mRNA codons with their cognate aminoacyl-tRNAs, containing the corresponding anticodons, during ribosomal protein synthesis (**Zaher and Green, 2009**; **Rodnina, 2012**). Cognate amino acids are attached to their respective tRNAs by aminoacyl-tRNA synthetases (aaRSs), and the ability of these enzymes to distinguish between cognate and non-cognate substrates is a major determinant of the fidelity of the genetic code. AaRSs discriminate against near- and non-cognate tRNAs at levels compatible with typical translation error rates ($\sim 10^{-4}$) due to the structural complexity and diversity observed between tRNA isoacceptors. AaRSs can less successfully discriminate against near-cognate amino acids, which may differ from the cognate substrate by as little as a single methyl or hydroxyl group. Errors during amino acid recognition do not usually compromise the accuracy of translation due to highly specific aaRS enzymes, and the widespread existence of editing mechanisms that proofread non-cognate amino acids. For example, phenylalanine tRNA synthetase (PheRS) edits mischarged Tyr-tRNA$^{Phe}$ at a hydrolytic editing site ~30 Å from the synthetic active site (**Roy et al., 2004**; **Kotik-Kogan et al., 2005**). PheRS editing provides a key checkpoint in quality control, as mischarged Tyr-tRNA$^{Phe}$ is readily delivered to the ribosome by

**eLife digest** Proteins are built from molecules called amino acids. The amino acids that make up a particular protein, and the order they appear in, are determined by the gene that encodes that protein. First, the gene is transcribed to produce a molecule of messenger RNA, which is then translated by a molecular machine called a ribosome. This involves other RNA molecules, called transfer RNAs (tRNAs), bringing the correct amino acids to the ribosome, which then joins the amino acids together to build the protein.

Amino acids are loaded onto their corresponding tRNA molecules by enzymes called tRNA synthetases. Occasionally, however, the wrong amino acid can be loaded onto a tRNA. If this amino acid ends up in a protein, the protein might not be able to function properly, or it might even be toxic to the cell, so cells need to be able to fix this problem. Some tRNA synthetases can check if a wrong amino acid has been loaded onto a tRNA, and remove it before it can cause harm. However, the importance of these 'editing' activities to living cells is unclear.

Here, Bullwinkle, Reynolds et al. show that, in the bacterium *E. coli*, a tRNA synthetase works to stop an incorrect amino acid—which accumulates in cells that are exposed to harmful chemicals—from being built into proteins. Without the enzyme's editing activity, the build-up of this amino acid slows the growth of the bacteria. However, *E. coli* can thrive without this editing activity when it is grown under normal conditions in a laboratory. Yeast benefit slightly from this editing activity when exposed to the stress-produced amino acid. But, unlike *E. coli*, yeast strongly rely on this activity when grown in an excess of another amino acid, which is used to build proteins but is the wrong amino acid for this tRNA synthetase.

The findings of Bullwinkle, Reynolds et al. will help to improve our understanding of which activities in a cell are most affected by mistakes in protein synthesis, and how these mistakes may relate to disease.

EF-Tu where it can efficiently decode Phe codons as Tyr in the growing polypeptide chain, resulting in mistranslation (*Ling et al., 2007b*, *2009*).

Despite their role in accurately translating the genetic code, aaRS editing pathways are not conserved, and their activities have varying effects on cell viability (*Bacher et al., 2005*; *Lee et al., 2006*; *Bacher and Schimmel, 2007*; *Bacher et al., 2007*). *Mycoplasma mobile*, for example, tolerates relatively high error rates during translation and lacks PheRS editing function, as do other aaRSs in this organism (*Li et al., 2011*; *Yadavalli and Ibba, 2013*). *Saccharomyces cerevisiae* cytoplasmic PheRS (*Sc*ctPheRS) has a low Phe/Tyr specificity and is capable of editing, whereas the yeast mitochondrial enzyme (*Sc*mtPheRS) completely lacks an editing domain, and instead relies on high Phe/Tyr specificity. *Escherichia coli,* in contrast, has retained both features and displays a high degree of Phe/Tyr specificity and robust editing activity (*Reynolds et al., 2010*). The range of divergent mechanisms used by different PheRSs to discriminate against non-cognate amino acids illustrates how the requirements for translation quality control vary with cellular physiology (*Yadavalli and Ibba, 2013*). Furthermore, given that editing by PheRS and other aaRSs is not essential for viability in yeast or *E. coli*, it is clear that the true roles of these quality control pathways remain to be fully elucidated (*Reynolds et al., 2010*).

In addition to the well-documented ability of aaRSs to edit tRNAs charged with genetically encoded near cognate amino acids, these same proofreading activities have been demonstrated to act on other non-canonical substrates. AaRSs are able to edit tRNAs misacylated with a range of amino acids not found in the genetic code such as homocysteine, norleucine, α-aminobutyrate and *meta*-tyrosine (*m*-Tyr), although the physiological relevance of these activities is unknown (reviewed in *Yadavalli and Ibba, 2012*). Both *E. coli* and *Thermus thermophilus* PheRS have been shown to edit *m*-Tyr, a metabolic byproduct formed by oxidation of phenylalanine following metal-catalyzed formation of hydroxyl radical species (*Huggins et al., 1993*; *Stadtman and Levine, 2003*; *Klipcan et al., 2009*). Certain species of fescue grasses (*Festuca* spp.) produce *m*-Tyr as a natural defense agent that appears in the proteomes of neighboring plants, and *m*-Tyr accumulation in the proteome of Chinese hamster ovary (CHO) cells has been proposed to have a cytotoxic effect on translation (*Gurer-Orhan et al., 2006*; *Bertin et al., 2007*). Taken together, these findings suggest that oxidative stress could potentially result in *m*-Tyr accumulation with the accompanying threat of cytotoxic mistranslation. Under such growth conditions,

the ability of the cell to edit $m$-Tyr-tRNA[Phe] would be essential to maintain cellular viability. Here we show that bacterial PheRS is able to efficiently edit $m$-Tyr-tRNA[Phe], and that this editing activity is essential for cellular growth and survival under both cytotoxic amino acid and oxidative stress conditions. Additionally, we show that PheRS editing in yeast provides only limited protection from $m$-Tyr, but instead is essential for protecting the cell from $para$-Tyr-tRNA[Phe] accumulation.

## Results

### PheRS editing is dispensable for *E. coli* and *S. cerevisiae* growth

To investigate the role of *E. coli* PheRS (*Ec*PheRS) editing in vivo, a strain was constructed containing a point mutation (G318W) within *pheT*, which encodes the β subunit of PheRS. Changes to residue βG318 hinder access to the editing site and thereby reduce *Ec*PheRS posttransfer editing activity by more than 70-fold in vitro (*Roy et al., 2004*; *Ling et al., 2007a*). *E. coli* strain NP37, which encodes a temperature sensitive *pheS* allele, was used as the background strain in order to facilitate selection of recombinant strains (*Kast et al., 1992*). Cell-free extracts from non-temperature-sensitive NP37-derived strains with wild type *pheT* and *pheT(G318W)* alleles were prepared and their PheRS activities tested. Only the strain encoding wild type PheRS retained post–transfer editing activity against $p$-Tyr-tRNA[Phe] (*Figure 1A*). Both strains showed identical levels of aminoacylation activity and growth at 37°C, indicating that the proofreading pathway is not required for growth under normal laboratory conditions. The role of PheRS editing was also investigated in *S. cerevisiae* by mutation of the chromosomal *FRS1* gene, which encodes the β-subunit of cytoplasmic PheRS (*Sc*ctPheRS). Introduction in *FRS1* of a mutation encoding the amino acid replacement D243A eliminated $p$-Tyr-tRNA[Phe] editing in vivo (*Figure 1B*; *Reynolds et al., 2010*) and had no effect on growth compared to wild type under standard conditions.

### PheRS editing specifies $m$-Tyr resistance in *E. coli*

Phenotypic microarrays (Biolog) were used to compare the growth of *E. coli pheT(G318W)* to wild type under 1920 growth conditions, and no significant changes were observed in the absence of PheRS editing. Additional experiments to investigate possible roles for editing under a range of other conditions, including heat shock, cold shock, pH stress and aging, failed to reveal differences compared to wild type. Growth of these strains was also compared in media containing varying concentrations of near-cognate $p$-Tyr in order to test the limits of *Ec*PheRS specificity in the absence of post–transfer editing

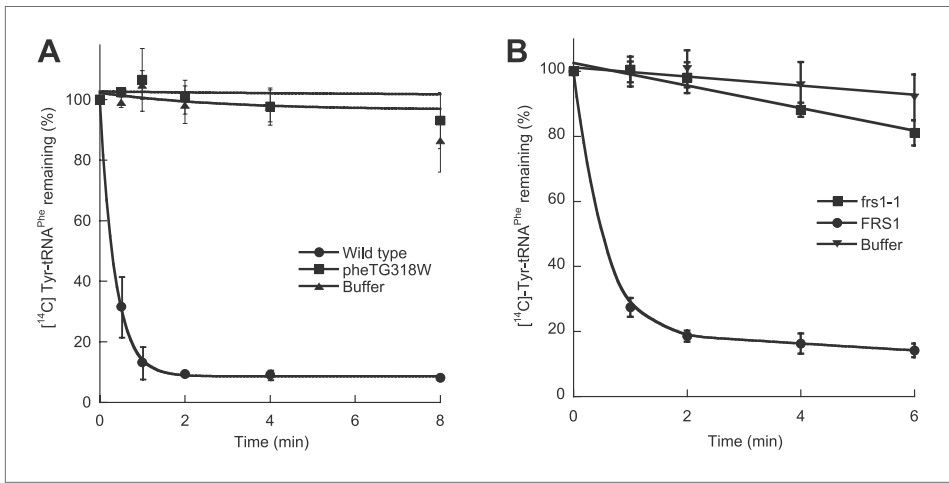

**Figure 1**. Chromosomal editing mutants of *E. coli* and *S. cerevisiae*. (**A**) Post–transfer hydrolysis of [14C]- Tyr-tRNA[Phe] (1 μM) by cell-free extracts isolated from wild type (●) and *pheT(G318W)* (■) *E. coli* strains (140 mg/ml total protein concentration) or buffer (▲) at 37°C. (**B**) Posttransfer editing activity of βD243A ctPheRS in *S. cerevisiae*. Reactions were performed at 37°C with 2 μM Tyr-tRNA[Phe] and *S. cerevisiae wild type* FRS1 or *frs1-1* (D243A) cell-free extracts normalized to aminoacylation activity (*Reynolds et al., 2010*). Data points are the mean of at least three independent experiments, with errors bars representing ±1 SD.

activity. Elevated concentrations of *p*-Tyr (>3 mM) did not affect the growth of *E. coli pheT(G318W)* compared to wild type (***Figure 2A***). Analysis of amino acid pools extracted from representative cells showed *E. coli pheT(G318W)* contained similar intracellular concentrations of *p*-Tyr and Phe as the wild type strain, indicating the *pheT* mutation has no effect on amino acid uptake (***Table 1***). In the absence of amino acid supplementation, the intracellular Phe:*p*-Tyr ratios were 1:1, and rose to 1:9 upon addition of *p*-Tyr. The growth of *E. coli pheT(G318W)* in the presence of *m*-Tyr, a non-proteinogenic amino acid previously shown to be a substrate for bacterial PheRS, was then investigated (***Klipcan et al., 2009***). Relative to wild type, growth of *E. coli* strain *pheT(G318W)* was inhibited in the presence of elevated intracellular concentrations of *m*-Tyr suggesting PheRS proofreading activity is needed to clear mischarged *m*-Tyr-tRNA^Phe in vivo (***Table 1***; ***Figure 2B***). Editing assays performed in vitro confirmed that, as with *p*-Tyr, post–transfer editing of *m*-Tyr-tRNA^Phe by PheRS is ablated by the G318W mutation (***Figure 2—figure supplement 1***). The inhibitory effect of *m*-Tyr on growth in the absence of editing was also observed in *E. coli* mutants derived from strain MG1655 that, unlike the NP37 background, encodes an intact stringent response (***Figure 2C***). The *pheT* editing mutation was also constructed in the MG1655 background in order to confirm the *m*-Tyr growth phenotype was not specific to strains lacking the stringent response, where cells are unable to properly sense and respond to amino acid starvation. Growth of *E. coli pheT(G318W)* was also evaluated in the presence of *ortho*-tyrosine (*o*-Tyr) and 3,4-dihydroxy-L-phenylalanine (L-DOPA), oxidation products of Phe and *p*-Tyr, respectively (***Maskos et al., 1992***). Neither of these non-proteinogenic amino acids inhibited growth of wild type or the *pheT(G318W)* mutant *E. coli* strain (***Figure 2—figure supplement 2***).

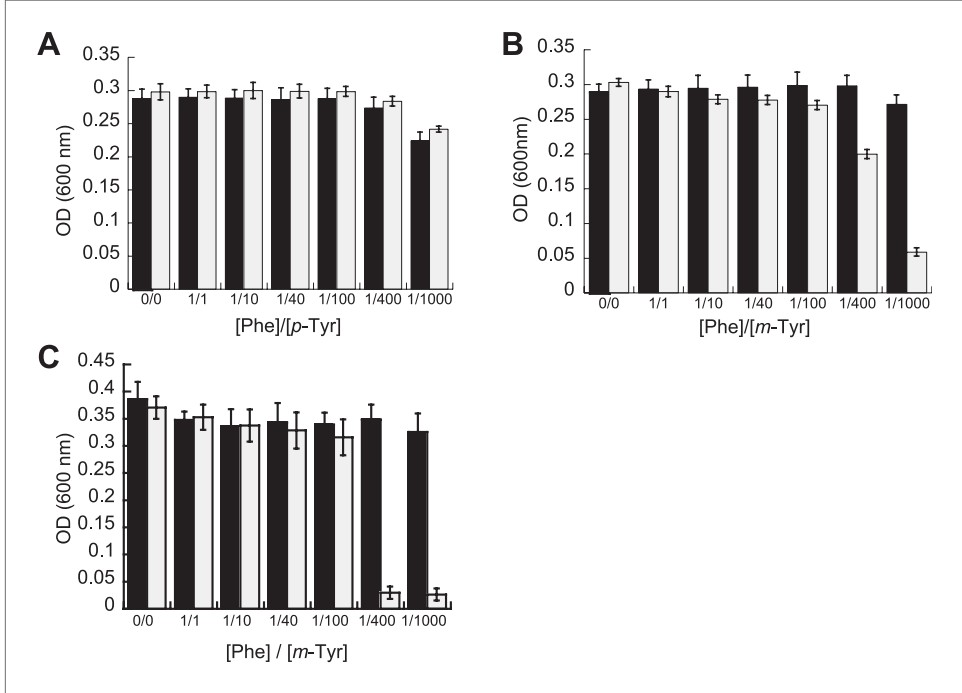

**Figure 2**. Effect of non-cognate amino acids on the growth of editing deficient *E. coli* strains. Growth of *E. coli pheT(G318W)* strain (grey bars) relative to wild type (black bars) under increasing concentrations of L-*p*-Tyr (**A**) or D,L-*m*-Tyr (**B**) relative to Phe. Cultures were grown in M9 minimal media supplemented with amino acids expressed as a ratio of Phe:Tyr. A ratio of 1:1 corresponds to 3 μM of each amino acid. (**C**) Growth of PheRS editing deficient strain of *E. coli* in an MG1655 background in the presence of different tyrosine isomers at 37°C. Bars are the mean of three independent cultures, with errors bars representing ± SD.
The following figure supplements are available for figure 2:

**Figure supplement 1**. *Ec*PheRS post–transfer editing of mischarged tRNA^Phe substrates.

**Figure supplement 2**. *E. coli* PheRS editing requirement for tyrosine isomers.

**Table 1.** Amino acid pools in wild type and editing defective *E. coli* strains

| Strain | Supplement | *m*-Tyr (µM)* | *p*-Tyr (µM) | Phe (µM) | *p*-Tyr/Phe | *m*-Tyr/Phe |
|---|---|---|---|---|---|---|
| Wild type | + *m*-Tyr | 2.9 ± 0.06 | 0.56 ± 0.1 | 0.63 ± 0.2 | 0.9 ± 0.0 | 5 ± 1 |
| *pheT(G318W)* | + *m*-Tyr | 2.7 ± 0.5 | 0.46 ± 0.02 | 0.90 ± 0.2 | 0.9 ± 0.2 | 6 ± 1 |
| Wild type | + *p*-Tyr | ND | 11 ± 4 | 0.91 ± 0.1 | 12 ± 4 | ND† |
| *pheT(G318W)* | + *p*-Tyr | ND | 8.9 ± 0.4 | 0.93 ± 0.1 | 9.7 ± 1 | ND |

*Concentrations of intracellular Phe and Tyr isomers isolated from wild type and *pheT(G318W) E. coli* strains grown in M9 minimal media supplemented with either *m*-Tyr or *p*-Tyr.
†ND indicates concentrations were below the detectable limit (0.01 µM).

The role of PheRS editing on yeast growth was tested under similar conditions to those examined for *E. coli*. While the editing deficient *frs1-1 (D243A)* yeast strain displayed no difference to wild type under heat shock or ethanol stress, it showed a pronounced defect in *p*-Tyr resistance. At elevated *p*-Tyr concentrations, growth of the *frs1-1 (D243A)* strain was restricted compared to wild type (**Figure 3A**), while the growth of both strains was more comparably inhibited by addition of *m*-Tyr (**Figure 3B**). These findings are in contrast to the responses of *E. coli* to tyrosine isomer stresses, consistent with the comparatively low Phe/*p*-Tyr amino acid specificity of the yeast enzyme and the previously observed inability of eukaryotic cytoplasmic PheRS to efficiently edit *m*-Tyr-tRNA^Phe (**Klipcan et al., 2009**; **Reynolds et al., 2010**).

## Bacterial and eukaryotic PheRSs have divergent tyrosine isomer specificities

*E. coli* PheRS is able to edit preformed *m*-Tyr-tRNA^Phe (**Klipcan et al., 2009**), and the loss of this activity in the G318W variant indicates that editing occurs at the site previously described for *p*-Tyr-tRNA^Phe (**Ling et al., 2007a**; **Figure 2—figure supplement 1**). Wild type *Ec*PheRS did not stably charge tRNA^Phe with either *m*- or *p*-Tyr, while G318W utilized both isomers for aminoacylation, with *m*-tyr being a more efficient substrate (**Figure 4A,B**). Under similar conditions, G318W PheRS was unable to utilize *o*-Tyr or L-DOPA for tRNA^Phe aminoacylation, consistent with the absence of any growth phenotype of the *pheT(G318W)* strain in the presence of these tyrosine analogs (**Figure 2—figure supplement 2**). As a substrate for *T. thermophilus* PheRS, L-DOPA has been shown to be 1500-fold less efficient than Phe (**Moor et al., 2011**). Examination of amino acid substrate specificity showed the catalytic efficiency ($k_{cat}/K_M$) for *m*-Tyr activation by *Ec*PheRS to be 35-fold less than for Phe, in contrast to *p*-Tyr which is activated almost 3000-fold less efficiently than the cognate substrate (**Table 2**). The ability of *Ec*PheRS

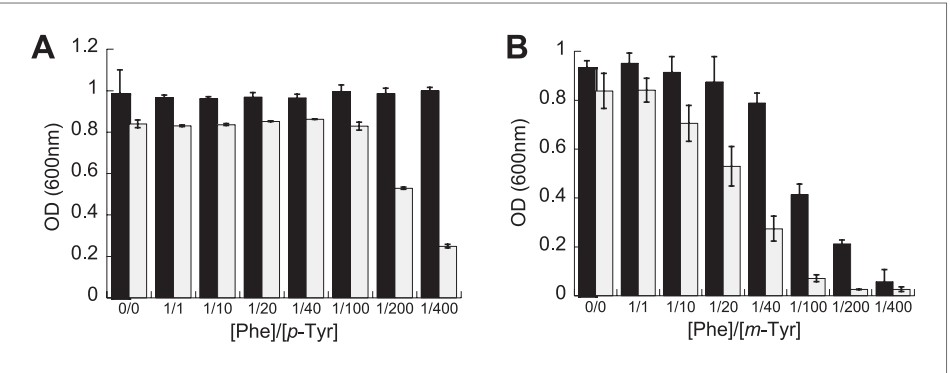

**Figure 3**. Effect of non-cognate amino acids on the growth of an editing deficient *S. cerevisiae* strain. Growth of yeast *frs1-1* (D243A) strain (grey bars) relative to a wild type strain (black bars) under increasing concentrations of L-*p*-Tyr (**A**) or D,L-*m*-Tyr (**B**) relative to Phe. Cultures were grown in minimal media supplemented with amino acids expressed as a ratio of Phe:Tyr. A ratio of 1:1 corresponds to 3 µM of each amino acid. Data points are the mean of three independent cultures, with errors bars representing ±1 SD.

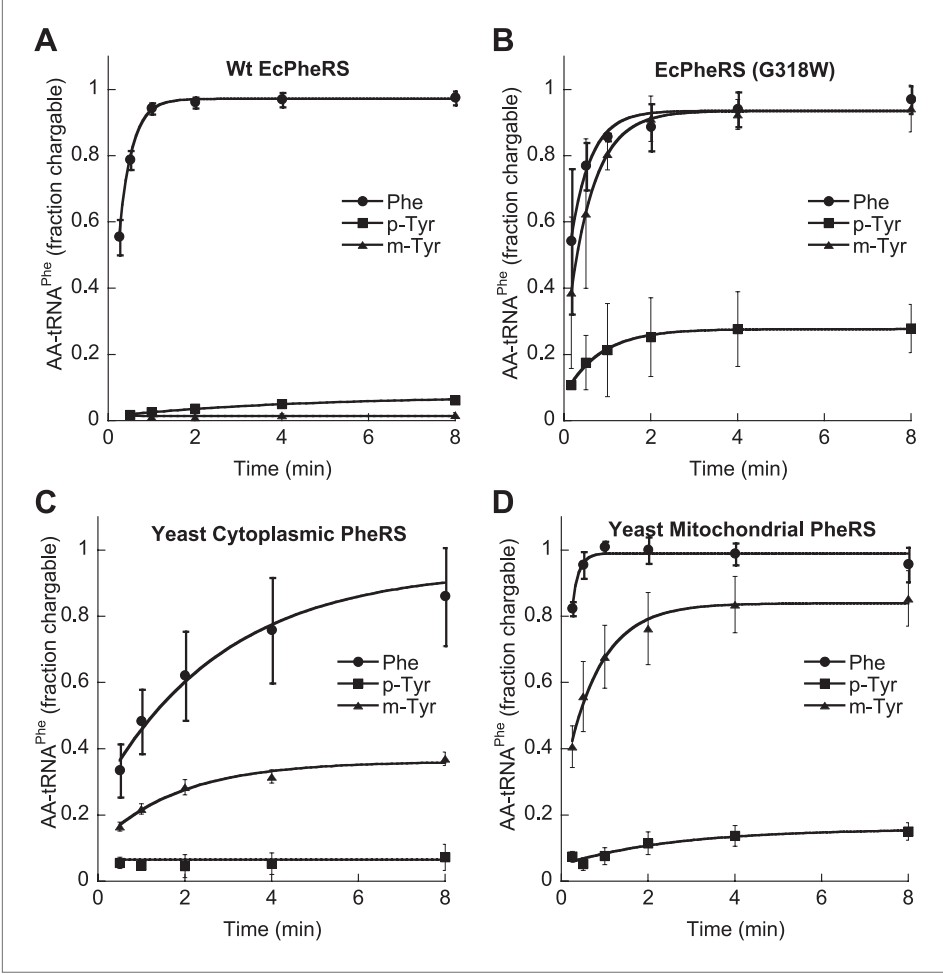

**Figure 4**. Tyrosine isomers as substrates for tRNA$^{Phe}$ aminoacylation by PheRS variants. tRNA$^{Phe}$ aminoacylation activities of (**A**) wild type and (**B**) G318W *E. coli* PheRS for 60 μM cognate Phe and non-cognate *p*- and *m*-Tyr substrates. Aminoacylation activities of (**C**) wild type cytoplasmic and (**D**) wild type mitochondrial *S. cerevisiae* PheRS for 100 μM cognate Phe and non-cognate *p*- and *m*-Tyr substrates. Data points are the mean of three independent experiments, with errors bars representing ± SD.

to efficiently activate *m*-Tyr is consistent with the need for editing to maintain cellular viability during growth in the presence of this non-proteinogenic amino acid.

In contrast to the *E. coli* enzyme, wild type *Sc*ctPheRS efficiently utilizes *m*-Tyr for activation and aminoacylation of tRNA$^{Phe}$. Charging of tRNA$^{Phe}$ with *m*-Tyr was seen at amino acid substrate concentrations where *p*-Tyr-tRNA$^{Phe}$ synthesis was not detected (***Figure 4C***; ***Table 2***). The $k_{cat}/K_M$ of *m*-Tyr activation by *Sc*ctPheRS is 71-fold lower than that of Phe, demonstrating relatively poor discrimination between the two amino acids (***Table 2***). In contrast to the *E. coli* enzyme, *p*-Tyr-tRNA$^{Phe}$ is a better

**Table 2.** Steady-state kinetic constants for amino acid activation by PheRS from *E. coli* and *S. cerevisiae* cytoplasmic PheRS

| PheRS | Phe | | | *m*-Tyr | | | *p*-Tyr | Specificity ($k_{cat}/K_M/k_{cat}/K_M$) | |
| | $K_M$ (μM) | $k_{cat}$ (s$^{-1}$) | $k_{cat}/K_M$ (s$^{-1}$/μM) | $K_M$ (μM) | $k_{cat}$ (s$^{-1}$) | $k_{cat}/K_M$ (s$^{-1}$/μM) | $k_{cat}/K_M$ (s$^{-1}$/μM) | Phe/*m*-Tyr | Phe/*p*-Tyr |
|---|---|---|---|---|---|---|---|---|---|
| *E. coli* | 18 ± 4 | 5.2 ± 2 | 0.29 | 247 ± 60 | 2.1 ± 0.8 | 0.008 | 1.1 × 10-4 | 35 | 2650 |
| Yeast ct | 16 ± 2 | 26 ± 4 | 1.6 | 1150 ± 230 | 26 ± 4 | 0.023 | 0.014 | 71 | 120 |

substrate for post–transfer editing by *Scc*tPheRS relative to *m*-Tyr-tRNA$^{Phe}$ (*Figure 5*). These results provide a possible explanation for the toxic effects *m*-Tyr has on the wild type yeast strain (*Figure 3B*), although additional cytotoxic affects of *m*-Tyr outside of translation cannot be ruled out. Post–transfer editing of *m*-Tyr-tRNA$^{Phe}$ by *Scc*tPheRS provides some protection from *m*-Tyr's toxic affects as there is a difference in the growth of wild type vs the *frs1-1(D243A)* strain at high concentrations of *m*-Tyr (*Figure 3B*). The mitochondrial variant of yeast PheRS (*Scm*tPheRS), which naturally lacks Tyr-tRNA$^{Phe}$ post–transfer editing activity (*Roy et al., 2005*), was also found to synthesize *m*-Tyr-tRNA$^{Phe}$ more efficiently than *p*-Tyr-tRNA$^{Phe}$ at similar tyrosine isomer concentrations (*Figure 4D*). The absence in yeast of appropriate quality control pathways in either the cytoplasm or mitochondria suggests that *m*-Tyr toxicity results from the accumulation of mischarged tRNAs in both compartments.

## *m*-Tyr is incorporated into the *E. coli* proteome at Phe codons

The correlation between *E. coli* PheRS-dependent *m*-Tyr toxicity in vivo and synthesis of *m*-Tyr-tRNA$^{Phe}$ in vitro strongly suggests that this mischarged tRNA is a substrate for ribosomal peptide synthesis. Dipeptide synthesis was monitored in vitro using *m*-Tyr-tRNA$^{Phe}$:EF-Tu:GTP as a substrate for decoding of a ribosomal A site Phe (UUC) codon. Under these conditions similar levels of fMet-*m*-Tyr and fMet-Phe were synthesized, indicating a lack of discrimination against the non-proteinogenic amino acid at the A-site of *E. coli* ribosomes (*Figure 6A*).

The effect of *m*-Tyr on protein synthesis in vivo was investigated by analyzing the accumulation of the non-proteinogenic amino acid in the proteomes of wild type and *E. coli pheT(G318W)* cells. Cytosolic protein samples were isolated from *m*-Tyr treated *E. coli* cells and samples subjected to acid hydrolysis to generate individual amino acids. The resulting amino acid hydrolysate was analyzed by liquid chromatography tandem mass spectrometry with multiple reaction monitoring (LC-MS/MS-MRM). To validate peak assignments of the Tyr isomers, co-chromatography was performed with synthetic *m*-Tyr or *o*-Tyr added to proteome samples. Only one peak for each of the isomers was observed, validating the assignments. Some level of *m*-Tyr was found to be present in the proteomes of both wild type and *phe(G318W)* strains indicating incorporation could be occurring through more than one route. Comparison of proteome total amino acid levels between wild type and *pheT*(G318W) strains indicated a level of misincorporation of 1% *m*-Tyr at Phe codons due to the absence of PheRS editing (*Figure 6B*). In wild type proteins the fraction of *m*-Tyr compared to Phe is 0.015, increasing to 0.025 in samples isolated from the *pheT(G318W)* strain grown in the same conditions. This result indicates post–transfer editing by PheRS provides protection of the *E. coli* proteome from misincorporation of *m*-Tyr at Phe codons. Quantification of *p*-Tyr relative to Phe in the protein samples isolated from cultures grown in the presence of 0.5 mM *p*-Tyr does not change between the wild type and *pheT(G318W)*

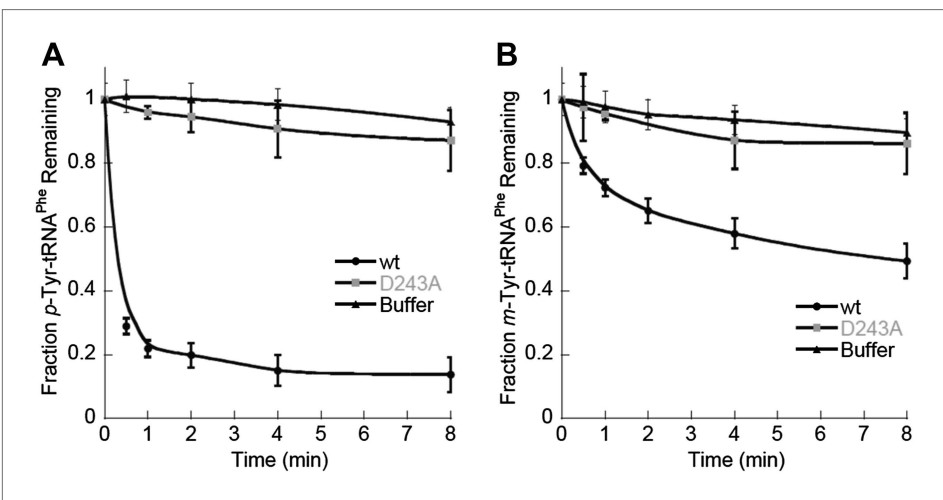

**Figure 5**. *Scc*tPheRS post–transfer editing of mischarged tRNA$^{Phe}$ substrates. Hydrolysis of 0.1 μM yeast (**A**) *p*-Tyr-[$^{32}$P]-tRNA$^{Phe}$ or (**B**) *m*-Tyr-[$^{32}$P]-tRNA$^{Phe}$ in the presence of 10 nM wild type *Scc*tPheRS (●) D243A *Scc*tPheRS (■) or buffer (▲) at 37°C. Data points are the mean of three independent experiments, with errors bars representing ± SD.

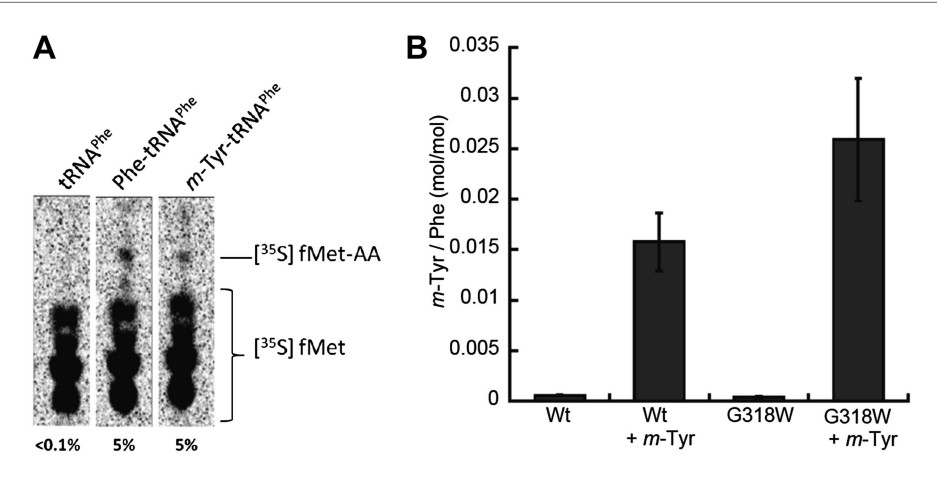

**Figure 6**. Incorporation of *m*-Tyr into the proteome of *E. coli*. (**A**) *In vitro* 70S ribosomal di-peptide synthesis with either Phe-tRNA^Phe or *m*-Tyr-tRNA^Phe (**B**) LC-MS/MS-MRM quantification of *m*-Tyr and Phe in protein hydrolysis isolated from *E. coli* expressed as molar ratio of *m*-Tyr to Phe. Wild type (Wt) and *pheT*(G318W) strains grown in M9 minimal media alone and supplemented with *m*-Tyr are shown. Error bars represent ± standard error of means.

The following figure supplements are available for figure 6:

**Figure supplement 1**. *p*-Tyr is not misincorported in the proteome of *E. coli* at Phe codons.

**Figure supplement 2**. *E. coli* TyrRS uses m-Tyr.

strains indicating this protein amino acid is not significantly misincorporated at Phe codons, even in the absence of PheRS editing (*Figure 6—figure supplement 1*). These analyses show a ratio of *p*-Tyr/Phe of 0.6, which correlates reasonably well with previous estimates of amino acid usage in *E. coli* (0.7, *Jauregui et al., 2000*).

A detectable level of *m*-Tyr in the proteome of wild type *E. coli* suggests either this non-proteinogenic amino acid escapes PheRS editing, infiltrates the proteome by means other than misincorporation at Phe codons or is carried over during cytosolic protein preparation. To measure the approximate amount of carryover, wild type PheRS *E. coli* strain was grown in the presence of 0.5 mM *o*-Tyr, which is not a substrate for protein synthesis, and total protein samples were subjected to acid hydrolysis and LC-MS/MS-MRM. In these samples, traces of *o*-Tyr were detected, indicating that free amino acid carry over possibly contributes to some of the *m*-Tyr detected in the samples from the wild type strain grown in M9 minimal media supplemented with *m*-Tyr. Whether the *m*-Tyr seen in the proteome of *E. coli* containing PheRS editing is formed post-translationally or is incorporated during protein synthesis via another promiscuous tRNA synthetase in *E. coli* is unclear. Aminoacylation of tRNA^Tyr with *m*-Tyr by *E. coli* TyrRS was detected in vitro, suggesting this synthetase may provide a route of *m*-Tyr incorporation even when PheRS editing is active (*Figure 6—figure supplement 2*).

## *E. coli* PheRS editing is required for growth under oxidative stress conditions

Reactive oxygen species (ROS) generated under oxidative stress via the Fenton reaction are capable of catalyzing the conversion of Phe to *m*-Tyr, which could potentially threaten the fidelity of protein synthesis in the absence of editing (*Maskos et al., 1992*; *Stadtman and Levine, 2003*). To investigate if oxidative stress conditions generate potentially toxic levels of *m*-Tyr in vivo, wild type and editing deficient *E. coli* were grown in the presence of $H_2O_2$ and $FeSO_4$ ($Fe^{2+}$) as a source of ROS. LC-MS/MS-MRM analyses showed that *m*-Tyr accumulated in the intracellular amino acid pools of ROS-treated cells (*Figure 7A*). In addition to *m*-Tyr, significant de novo *o*-Tyr accumulation was also observed following ROS treatment, although this is not expected to pose a threat to translation fidelity as it is not a substrate for PheRS (*Figure 2—figure supplement 2*). *E. coli* lacking PheRS editing activity showed a reduction in growth relative to wild type when grown in media where ROS exposure increased,

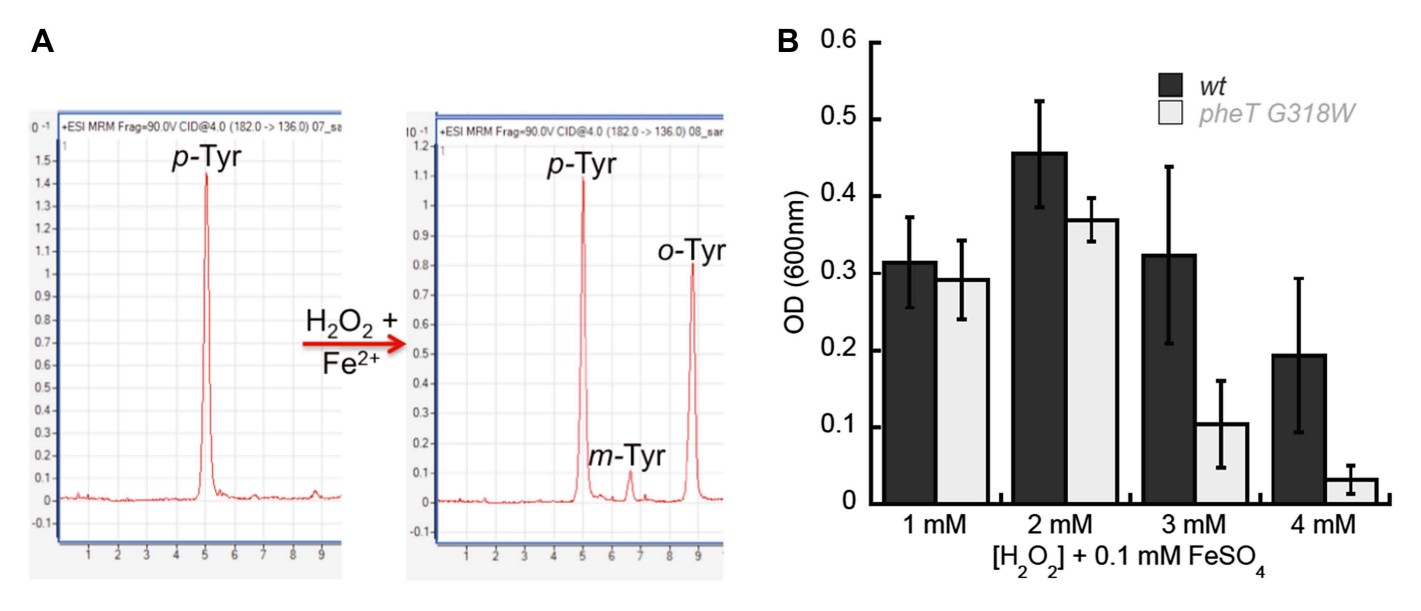

**Figure 7**. Requirement for PheRS posttransfer editing in ROS conditions in vivo. (**A**) LC-MS/MS-MRM chromatograms for $p$-, $m$- and $o$-Tyr (m/z 182→136 transition) extracted from cells grown in the absence (left) and presence (right) of $H_2O_2$ and $FeSO_4$. (**B**) Growth of *E. coli pheT(G318W)* strain relative to wild type in M9 minimal media supplemented with 0.1 mM $FeSO_4$ and increasing concentrations of $H_2O_2$. Bars are the mean of three independent cultures, with errors bars representing ± SD.

consistent with the accumulation of free $m$-Tyr and its subsequent utilization in protein synthesis (*Figure 7B*). Taken together, our data indicate that PheRS editing activity affords *E. coli* protection against the co-translational insertion of non-proteinogenic amino acids that accumulate during oxidative stress. Attempts to identify $m$-Tyr in the total protein hydrolysis samples under oxidative stress conditions revealed the presence of $m$-Tyr and $o$-Tyr in both the wild type and *pheT(G318W)* strains. Proper quantification of the levels of each amino acid in these samples was not possible as adequate resolution could not be achieved for the peaks corresponding to the different Tyr isomers in total protein samples prepared from $H_2O_2$ treated cells. These observations suggest posttranslational damage of Phe residues in protein by $H_2O_2$ treatment may also be partially responsible for the accumulation of hydroxylated Phe residues. In efforts to increase the misincorporation of $m$-Tyr into the proteome at Phe codons, higher levels of $H_2O_2$ were used, however this resulted in the death of both strains likely due to the other damaging effects of reactive oxygen species.

## Discussion

### Context dependent specificity and editing

It has long been proposed that the fidelity of aminoacyl-tRNA synthetases needs to be at or above 1 in 3,000, which is cited as an approximate overall level of error for protein synthesis (*Loftfield and Vanderjagt, 1972*). AaRS fidelity is achieved through discrimination at the aminoacylation site as well as through additional editing activities in some aaRSs. Protection against both $p$-Tyr and $m$-Tyr incorporation at Phe codons appears critical in *E. coli* as the PheRS enzyme maintains high active-site selectivity against $p$-Tyr as well as post–transfer editing activity against $m$-Tyr-tRNA$^{Phe}$. *E. coli* PheRS requires this editing activity to protect the proteome from toxic effects of the non-proteinogenic amino acid $m$-Tyr, which is poorly discriminated against by the active site of the enzyme. Examination of the structure of the catalytic active site provides clues as to why PheRS is unable to discriminate against all the Tyr isomers. Ala294 is primarily responsible for specificity against binding of *para*-substituted Phe analogs, while Gln174 and Glu210 help stabilize the hydroxyl of non-cognate $m$-Tyr at position 3 of the ring (*E. coli* numbering) (*Klipcan et al., 2009*). In the case of the cognate Phe substrate, Glu210 is also needed to hydrogen bond with the Phe amino group, ensuring correct orientation of the substrate for activation (*Safro et al., 2005*; *Mermershtain et al., 2011*). It is unlikely this enzyme selects against

recognition of *m*-Tyr while still maintaining efficient activity for the cognate amino acid, therefore the maintenance of post–transfer editing activity is critical for fidelity in *E. coli*. In eukaryotes, cytoplasmic PheRS editing is needed to protect the proteome from *p*-Tyr misincorporation. This finding concurs with the low Phe/*p*-Tyr specificity of the yeast cytoplasmic enzyme (*Reynolds et al., 2010*). It is unclear if protection from *m*-Tyr incorporation is achieved through editing as the yeast strain encoding wild type ctPheRS is sensitive to high concentrations of *m*-Tyr, mtPheRS efficiently aminoacylates *m*-Tyr onto tRNA^Phe, and other eukaryotic proteomes are vulnerable to the use of this oxygen-damaged amino acid for translation (*Gurer-Orhan et al., 2006*). Taken together, these findings suggest that either *m*-Tyr accumulation is not a substantial threat in eukaryotes, or possibly that the incorporation of low amounts of this non-proteinogenic amino acid in certain proteomes confers some as yet unknown evolutionary benefits.

## Non-proteinogenic amino acids as threats to translational integrity

Naturally occurring non-proteinogenic amino acids occur widely in nature and are well-characterized by-products and/or intermediates of biosynthetic pathways (*Richmond, 1962*). The actual threats these non-canonical substrates pose to protein synthesis and cell viability is unknown, as is the role of aaRS quality control in protecting the proteome from such amino acids. The non-proteinogenic amino acid *m*-Tyr has been detected in several eukaryotic proteomes and is one of the products of canonical aromatic amino acid oxidation (*Wells-Knecht et al., 1993*; *Pennathur et al., 2001*). The presence of hydroxylated forms of tyrosine in proteomes has previously been attributed to post-translational damage to proteins by hydroxyl radical species, and is often used as a marker for tissue damage due to the oxidative conditions of aging and disease. It has also been shown that *m*-Tyr and other Tyr analogs, for example L-DOPA, are substrates for translation in some organisms and could potentially be incorporated directly during protein synthesis (*Rodgers et al., 2002*; *Ozawa et al., 2005*; *Gurer-Orhan et al., 2006*; *Klipcan et al., 2009*; *Moor et al., 2011*). Our results now reveal the role of *E. coli* PheRS editing for preventing the use of *m*-Tyr during protein synthesis, demonstrating the threat amino acid oxidation poses to the proper functioning of the bacterial translation machinery.

Incorporation of *m*-Tyr into the proteome of *E. coli* at Phe codons is toxic to the cell, and in the absence of PheRS quality control this non-proteinogenic amino acid serves as an efficient substrate for translation. Other non-proteinogenic amino acids have also been shown to be potential threats to translation, such as α-aminobutyrate, which in the absence of ValRS editing is toxic at high concentrations, although the physiological conditions under which this non-protein amino acid might naturally accumulate to significant levels are unclear (*Nangle et al., 2002*). The robust editing activity maintained by *E. coli* PheRS to protect the proteome from *m*-Tyr demonstrates the significant threat such an amino acid poses when misincorporated at specific near-cognate codons. In contrast, the presence of *m*-Tyr in the proteome of wild type *E. coli* suggests misincorporation can also occur at Tyr codons but without cytotoxic sequelae. These findings suggest that the cellular effects of non-proteinogenic amino acid incorporation are codon-dependent. The cell does not have codons or tRNAs for *m*-Tyr, therefore any advantage or disadvantage this amino acid might provide to the proteome cannot easily be selected for, or against, at the level of the genetic code. The only selection against near-cognate non-proteinogenic amino acid use during translation can be made at the level of the synthetase or ternary complex formation with an elongation factor (*Dale and Uhlenbeck, 2005*). In *E. coli*, misincorporation of *m*-Tyr at Phe codons in the absence of PheRS quality control occurred at a frequency of 1% and had a significant impact on cellular viability and restricted growth. This contrasts with the effects of misincorporation of canonical amino acids, which have been shown to be tolerated at rates of up to 10% without inhibiting growth (*Ruan et al., 2008*). Taken together with earlier studies, our present findings now show that the misincorporation of non-proteinogenic amino acids presents a substantial challenge for protein synthesis quality control. This in turn suggests that many 'dispensable' editing functions, both in aaRSs and *trans* editing factors, may actually be essential for growth under conditions that lead to the accumulation of potentially toxic levels of non-proteinogenic amino acids.

## Oxidative stress and translation quality control

Oxidation of amino acids by reactive species such as hydroxyl radical and superoxide anions results in limited alteration of amino acid structure, such as the addition of a hydroxyl group, creating potential in vivo substrates for tRNA misacylation. These damaged amino acids challenge the protein synthesis machinery, as for example in the case of L-DOPA, and leucine hydroxide, which have been shown to

be incorporated into proteins in mouse cells (**Rodgers et al., 2002**; **Ozawa et al., 2005**). The formation of intracellular $m$-Tyr in *E. coli* under physiological conditions is possibly a result of cellular exposure to $H_2O_2$. Aerobic respiration results in elevated endogenous levels of $H_2O_2$, but bacterial cells are also exposed to ROS present in their environment. Uncharged $H_2O_2$ is able to penetrate the cell membrane and accumulate inside cells whenever $H_2O_2$ is present in the extracellular habitat. At physiological pH, $H_2O_2$ quickly oxidizes ferrous iron via the Fenton reaction, generating a hydroxyl radical that can react with nearby cellular targets (**Rush et al., 1990**). Accumulation of toxic levels of $m$-Tyr in the intracellular pools of *E. coli* occurs under experimental conditions that promote the formation of hydroxyl radicals, however this is not the major byproduct of Phe oxidation. The $o$-Tyr isomer is the more abundant hydroxylation product under ROS-generating conditions used here (**Figure 7**). However, there is no observed $o$-Tyr aminoacylation of tRNA$^{Phe}$ by wild type PheRS in vitro, or inhibition of cell growth, in the presence of this hydroxylated Phe substrate. These, and the corresponding biochemical data, indicate how *E. coli* PheRS has evolved to effectively discriminate against different Tyr isomers using a combination of substrate specificity ($o$-Tyr, $p$-Tyr) and editing activity ($m$-Tyr, $p$-Tyr). In contrast, yeast PheRS has mainly evolved specifically to discriminate for $p$-Tyr by editing, reflecting differences in the factors that drive selection of quality control mechanisms.

## Materials and methods

### Strains, plasmids, and general methods

Proteins and tRNAs were prepared essentially as described previously (**Roy et al., 2005**). Mutation of the *E. coli* PheRS gene in the pQE31-EcFRS expression plasmid was completed using standard polymerase chain reaction (PCR)-based site-directed mutagenesis as previously described (**Ling et al., 2007b**). Purification of His-tagged PheRS variants included dialysis against two changes of 25 mM Tris–HCl pH 7.5, 100 mM KCl, 0.1 mM sodium pyrophosphate, 3 mM 2-mercaptoethanol, and 10% glycerol, in order to release any enzyme-bound adenylate. Dialysis against 25 mM Tris–HCl pH 7.5, 100 mM KCl, 5 mM $MgCl_2$, 3 mM 2-mercaptoethanol, and 10% glycerol was then performed followed by dialysis against this same buffer with 50% glycerol, flash frozen, and stored at −80°C. Active enzyme concentration was determined by active site titration as previously described (**Ibba et al., 1994**). Phenylalanine, L-$p$-tyrosine and D,L-$m$-tyrosine were purchased from Sigma-Aldrich (St. Louis, MO).

### Construction of editing defective *E. coli* and yeast mutant strains

The editing deficient strain of *E. coli*, *pheT(G318W)*, was constructed using established recombineering methods involving the lambda red/gam pKD46 plasmid (**Datsenko and Wanner, 2000**). The *pheS*$^{ts}$ *E. coli* strain NP37, which contains a G98D mutation (**Kast et al., 1992**) was used as the parental strain to allowed for selection of recombination events within the region of the neighboring *pheS* and *pheT* genes. Site directed mutagenesis of the pQE31-EcFRS wt plasmid (**Ling et al., 2007b**) was used to construct pQE31-EcFRSG318W/V364V. Linear PCR products were amplified from this plasmid and introduced to the pKD46 containing NP37 parent strain via electroporation. Primers for PCR were as follows: p14 EcFRS: 5′-AACCATGTCACATCTCGC and P16AS EcFRS: 5′-CGTTGGTGATATCAATTACCGG. This linear DNA contains the wild type *pheS* gene to allow for colony selection at 42°C, the *pheT* gene containing a G318W mutation, and a silent V364V mutation that introduces a BamHI site for screening of colonies. Recombinant strains were confirmed with sequencing. A wild type *pheS/pheT* strain was also constructed in the same manner, but without changing the Gly residue at 318. The λ-red recombineering system was used to introduce the *pheT(G318W)* mutation into the *E. coli* MG1655 background. Competent cells were prepared as previously described (**Yu et al., 2000**) of an MG1655 derivative containing pSIM6, a plasmid that carries the λ-red system (**Datta et al., 2006**). These cells were transformed with a 70-mer oligonucleotide (5′- CACAACAAGGCGCTGGCGATGGG<u>A</u>GG<u>A</u>AT<u>A</u>TT <u>T</u>TGGGG<u>A</u>GAG<u>C</u>ATTC<u>A</u>GGCGTGAAT GACGAAACACAAA) that has several wobble mutations (underlined) on either side of the *pheT(G318W)* mutation (bolded). The wobble mutations serve to overwhelm the mismatch repair system (**Costantino and Court, 2003**). Positive clones were identified by colony PCR, with a primer that recognized the mutated sequence (5′-<u>A</u>GG<u>A</u>AT<u>A</u>T TTTGGGG<u>A</u>GAG<u>C</u>A<u>T</u>TC<u>A</u>) and a reverse primer 500-bp distant (5′-CCGATCAGGCGATCC AGTTTG), and subsequent DNA sequencing. One clone was chosen to serve as the intermediate strain and was subjected to a second round of recombineering, as indicated above, with an oligo (5′CACAA

CAAGGCGCTGGCGATGGGCGGCATCTTCTGGGGCG AACACTC TGGCGTGAATGACGAAACACAAA) to remove the wobble mutations and leave solely the *pheT(G318W)* mutation. The intermediate strain was also transformed with an oligo (5'-CACAACAAGGCGCTGGCGATGGGCGGCATCTTCGGTGG CGAACACTCTGGCGTGAATGACGAAACACAAA) that would revert the strain back to the wild type *pheT* sequence. This strain served as the wild type control strain in studies with the *pheT(G318W)* derivative of *E. coli* MG1655. Again, positive clones were screened by colony PCR (primer 5'-CGGCATC TTCTGG GGCGAACACTCT for *pheT(G318W)* and primer 5'-CGGCATCTTCGGTGGCGAACACTCT for wild type, both with the reverse primer indicated above) and DNA sequencing.

Strains derived from *S. cerevisiae* W303 (*MATa/MATα, ade2-1, his3-11,15, leu2-3,112, trp1-1, ura3-1, can1-100*) were used to construct chromosomal mutants of *FRS1*. A 2084 bp fragment of *frs1-1*, obtained through PCR of the plasmid pFL36-frs1-1 (*Reynolds et al., 2010*), was inserted into the integrative plasmid YIP5 (*Struhl et al., 1979*) at the EcoRI and NruI restriction sites by In-Fusion cloning (Clontech, Mountain View, CA), resulting in the plasmid YIP5-frs1-1. W303 (*MATa/MATα, ade2-1, his3-11,15, leu2-3,112, ura3-1, can1-100*) was transformed with YIP5-frs1-1 and insertion of the plasmid was selected for by growth on complete supplement media minus uracil (CSM-Ura; Sunrise Science Products, San Diego, CA). Recombinant strains were grown in YPDA at 30°C, shaking at 300 rpm, for 24 hr, and plated on YPDA. Crossovers were selected for by replica plating onto media containing 5-flouroorotic acid (5-FOA). *TRP1* prototroph strains were created through the PCR amplification of the *TRP1* locus from *S. cerevisiae* strain BY4743 (*MATa/MATα, his3Δ1/his3Δ1, leu2Δ0/leu2Δ0, lys2Δ0/LYS2, MET15/met15Δ0, ura3Δ0/ura3Δ0*) and the linear product used to transform the W303 (*MATa/MATα, ade2-1, his3-11,15, leu2-3,112, trp1-1, ura3-1, can1-100, FRS1/frs1-1,*) strain. *TRP1* recombinants were selected on synthetic complete minus tryptophan media. Haploids were obtained by sporulation, dissection onto YPDA, replica plated onto complete supplement media minus tryptophan, and tryptophan prototroph colonies selected. Haploids were screened for the presence of the *frs1-1* mutation, resulting in the strains NR1 (*MATa, ade2-1, his3-11,15, leu2-3,112, ura3-1, can1-100*) and NR2 (*MATa, ade2-1, his3-11,15, leu2-3,112, ura3-1, can1-100, frs1-1*).

## Growth assays

Single colonies of *E. coli*, wild type *pheT* or *pheT(G318W)*, were picked from LB plates, resuspended in sterile water and used to inoculate liquid culture at an initial $OD_{600}$ of 0.04. Cultures were grown in M9 media supplemented with glucose (2 g/l), thiamine (1 mg/l), $MgSO_4$ (1 mM), $CaCl_2$ (0.1 mM), and varying amounts of amino acids. Cultures were grown at 37°C in 250 μl volumes using 96-well plates for ease of titrating several amino acid concentrations. Phe was kept constant at 0.003 mM and L-Tyr or D,L-*m*-Tyr was varied from 0.003 mM to 3 mM. Optical densities at 600 nm ($OD_{600}$) were read using a xMark Microplate Absorbance Spectrophotometer (Bio-Rad Laboratories, Hercules, CA) after 12–18 hr of growth. Growth curves were performed in supplemented M9 media containing none or 0.5 mM *D,L-m*-Tyr, and 100 ml cultures were grown at shaking at 37°C. Growth experiments in the presence of oxidative stress agents were also set up in 96-well plates in M9 minimal media containing 0.5 mM Phe, 0.1 mM $FeSO_4$, and 2-4 mM $H_2O_2$. For all growth assays of the *S. cerevisiae* strains NR1 and NR2, cells were streaked on YPDA and incubated at 30°C. After approximately 72 hr single colonies were picked, resuspended in sterile water and used to inoculate liquid cultures to an initial $OD_{600}$ of 0.01. Microtitre growth assays were completed by inoculating 150 μl of MM (Difco yeast nitrogen base without amino acids, 0.002% adenine, 0.002% uracil, 0.002% L-histidine, 0.01% L-leucine, and 2% glucose) + Phe:Tyr (where Phe was kept constant at 0.003 mM and Try was varied from 0.003 mM to 1.2 mM) in a 96 well microtitre plate. Plates were incubated at 30°C and growth was measured after 16 hr by $OD_{600}$.

## tRNA preparation and $^{32}$P labeling

Purified native *E. coli* tRNA$^{Phe}$ was purchased from Chemical Block, Moscow, Russia. *S. cerevisiae* cytoplasmic and mitochondrial tRNA$^{Phe}$ were made from T7 runoff transcription as previously described (*Roy et al., 2005*; *Yadavalli and Ibba, 2012*). DNA template for tRNA transcription was generated from plasmids carrying tRNA genes (*Sampson and Uhlenbeck, 1988*) by PCR amplification and extended only to C75 to allow for $^{32}$P labeling of A76. After ethanol precipitation, tRNA transcripts were purified on denaturing 12% polyacrylamide gel and extracted by electrodialysis in 90 mM Tris-borate/2 mM ethylenediaminetetraacetic acid (EDTA) (pH 8.0). The tRNA was phenol and chloroform extracted, ethanol precipitated, dried and resuspended in diethylpyrocarbonate

(DEPC)-treated ddH$_2$O. Refolding was carried out by heating the tRNA at 70°C for 2 min, followed by the addition of 2 mM MgCl$_2$ and slow cooling to room temperature. tRNAs were $^{32}$P-labeled at A76 essentially as described previously (*Roy et al., 2005*). For *E. coli* tRNA$^{Phe}$ the CCA-3′-end was removed prior to labeling by treatment of 20 µM tRNA transcript with 100 µg/ml *Crotalus atrox* venom (Sigma-Aldrich) in a buffer containing 50 mM Na-Gly (pH 9.0) and 10 mM magnesium acetate. The reaction was incubated for 40 min at 21°C and phenol/chloroform extracted, ethanol precipitated, and desalted by gel filtration through a Sephadex G25 column (GE Healthcare Life Sciences, Pittsburgh, PA). The CCA-3′-end of the tRNA was reconstituted and radiolabeled using *E. coli* tRNA terminal nucleotidyl-transferase and [α-$^{32}$P] ATP as described (*Roy et al., 2005*). Yeast cytoplasmic and mitochondrial tRNA$^{Phe}$ C75 transcripts were labeled the same way, however CTP was excluded from the reaction mix. Samples were treated with one volume of phenol, and the tRNA was phenol/chloroform extracted and gel filtered twice through a G25 column.

## Aminoacylation and editing assays

Aminoacylation reactions were performed at 37°C in aminoacylation buffer (100 mM Na-Hepes pH 7.2, 30 mM KCl, 10 mM MgCl$_2$, 10 mM DTT) with 8 mM ATP, 60 (*E. coli*) or 100 µM (*S. cerevisiae*) cold amino acid, 0.5 µM $^{32}$P-tRNA. PheRS (100 nM) was added to initiate the reactions. Aliquots were removed at designated time points, treated with an equal volume of 0.5 M sodium acetate pH 4.2 and incubated for 30 min at room temperature with S1 RNase (Promega, Fitchburg, WI). The free [α-$^{32}$P]AMP and aminoacyl-[α-$^{32}$P]AMP were separated by thin layer chromatography on polyethyle-neimine cellulose (Sigma-Aldrich) in 100 mM ammonium acetate, 5% acetic acid and visualized as described previously (*Wolfson and Uhlenbeck, 2002*). Mischarging of *E. coli* tRNA$^{Phe}$ was performed at 37°C for 20 min in aminoacylation buffer with 8 mM ATP, 100 µM cold with L-*p*-Tyr or D,L-*m*-Tyr, 4 µM $^{32}$P-tRNA and 1 µM αA294G/βG318W PheRS (*Roy et al., 2004*). Reactions were stopped by the addition of 1 volume of phenol pH 4.5, and the aminoacylated tRNA was phenol/chloroform extracted and gel filtered twice through a G25 column pre-equilibrated with 5 mM sodium acetate pH 4.2. Editing assays were performed in aminoacylation buffer and contained 0.1 µM Tyr-[$^{32}$P] tRNA$^{Phe}$, and 10 nM G318W PheRS. Reactions were arrested at various time points and analyzed by TLC as described for the aminoacylation reactions (see above). Editing assays of the cell-free extracts were performed similarly, however mischarged [$^{14}$C]Tyr-tRNA$^{Phe}$ was formed (*Roy et al., 2005*), and 1 µM was used in reactions containing aminoacylation buffer, 2 mM ATP, and cell free extract that was normalized for aminoacylation activity.

## ATP/PPi exchange

ATP/PPi exchange assays were performed according to standard methods as previously described (*Roy et al., 2004*, *2005*). Reactions were carried out at 37°C in a medium containing 100 mM Na-Hepes (pH 7.2), 30 mM KCl, 10 mM MgCl$_2$, 2 mM NaF, 2 mM ATP, 2 mM [$^{32}$P]PP$_i$ (2 cpm/pmol), varying amounts of Phe (1-200 µM) and D,L-*m*-Tyr (20–2000 µM), and 40 nM *E. coli* PheRS, 100-150 nM yeast cytosolic enzyme. After 1–1.3 min, 25 µl of the reaction were removed and added to a solution containing 1% charcoal, 5.6% HClO$_4$, and 75 mM PPi. The charcoal-bound ATP was filtered through a 3 MM Whatman filter discs under vacuum and washed three times with 5 ml of water and once with 5 ml of ethanol. The filters were dried, and the radioactivity content was determined by liquid scintillation counting. We previously reported the activation specificity of Phe vs *p*-Tyr to be 7800 (*Reynolds et al., 2010*), however this discrepancy appears to be due to differences in enzyme-bound aminoacyl adenylate during protein purification affecting the measured active enzyme concentration. This problem was resolved here through extensive dialysis against PPi.

## Dipeptide synthesis

Initiation complexes (70S IC) were formed using tight coupled 70S ribosomes, [$^{35}$S]fMet-tRNA$^{fMet}$, Met–Phe coding mRNA, and initiation factors essentially as described (*Bullwinkle et al., 2013*). Ternary complexes were formed using aminoacylated tRNA$^{Phe}$ and activated EF-Tu (*Bullwinkle et al., 2013*). Reactions were initiated by mixing 1 µM ternary complex with 0.1 µM 70S IC and incubated for 1 min at 21°C before quenching with 1/5$^{th}$ volume of 2 M KOH and 1 M H$_2$O$_2$. Quenched reactions were then incubated at 37°C for 20 min to deacylate tRNA$^{Phe}$, and [$^{35}$S]fMet-Phe dipeptides were separated from [$^{35}$S]fMet by TLC on silica plates in buffer containing 1-butanol:acetic acid:H$_2$O (4:1:1). TLC plates were then exposed and quantified by phosphor imaging.

## Quantification of amino acid pools

Cultures were grown to late log phase in supplemented M9 media with or without 0.5 mM tyrosine in 5 ml volumes and harvested by vacuum filtration over a nylon filter followed by washing cells three times with 1 ml $H_2O$. Cells and filters were then placed upside down in 0.5 ml extraction buffer (40% acetonitrile, 40% methanol) containing internal standards (100 pmol [U$^{13}$C]Phe and 100 pmol [U$^{13}$C]Tyr) at −20°C for 15 min. Metabolites were extracted as described (*Crutchfield et al., 2010*) and vacuumed dried. Samples were re-dissolved in water (50 µl), centrifuged (16,000×$g$, 5 min) and the supernatant transferred to LC injector vials. Aliquots of the supernatant (typically 5 µl) were injected onto a reverse phase HPLC column (Phenomenex Kinetex XB-C18, 2.1 × 100 mm, 1.7 µm particle size, 100 Å pore size) equilibrated in solvent A (water/formic acid, 100/0.1, vol/vol) and eluted (100 µl/min) with an increasing concentration of solvent B (acetonitrile/formic acid, 100/0.1, vol/vol; min/%B, 0/1, 5/1, 26/70, 27/1, and 35/1). The effluent from the column was directly connected to an electrospray ion source (Agilent Jet Stream, Agilent, Santa Clara, CA) attached to a triple quadrupole mass spectrometer (6460; Agilent) scanning in the multiple reaction monitoring mode with standard resolution settings (FWHM 0.7) using previously optimized conditions for the following transitions: Tyr, 182→136; U$^{13}$C-Tyr, 191→144; Phe, 166→120; U$^{13}$C-Phe, 175→128. With each batch of samples a series of standards was prepared with the same amount of internal standards and increasing amounts of Tyr and Phe (0, 0.1, 1, 10 and 100 pmol in 50 µl of water, in duplicate). Typical retention times for *p*-Tyr, *m*-Tyr, *o*-Tyr and Phe were 4.8, 6.6, 8.6 and 9.7 min, respectively. Peak areas were measured using instrument manufacturer supplied software (Agilent MassHunter). The amount of each analyte in each sample was determined by interpolation from the curves constructed from the standard samples (peak area Tyr or Phe/peak area U$^{13}$C-Tyr or –Phe against amount of Tyr or Phe).

## Purification and LC-MS/MS-MRM of total protein hydrolysate

*E. coli* cultures (100 ml), prepared in duplicate, were grown in M9 minimal media with or without 0.5 mM *m*-Tyr to exponential phase and harvested by centrifugation (6000×$g$, 10 min). Cell pellets were washed twice, resuspended in water, and lysed by sonication. To precipitate ribosomes and nucleic acids, streptomycin sulfate was added to a final concentration of 8 mg/ml (*Cohen and Lichtenstein, 1960*). Samples were incubated at 4°C for one hour, then centrifuged at 11,000×$g$ for 5 min. Supernatants were collected and brought to 55% acetone by volume at 4°C for 1 hr. Precipitated material was pelleted at 11,000 g for 5 min. Supernatants were discarded and the pellets were washed twice with 60% acetone (ice cold). The pellets were then subjected to two methanol/chloroform extractions, vacuumed dried, and weighed. One set of samples was used for measurement of protein content (bicinchoninic assay, Thermo Scientific, Waltham, MA). After resuspending in water and addition of internal standards (U$^{13}$C-Tyr and U$^{13}$C-Phe, 100 pmol each), the other set of samples was subjected to acid hydrolysis (6 M HCl for 24 hr at 110°C). LC-MS/MS-MRM was performed on the hydrolysate as described above.

## Acknowledgements

Protein hydrolysis was performed with the help of John Lowenson and Steve Clarke (UCLA). We thank I Artsimovitch, K Musier-Forsyth and K Fredrick for invaluable discussions, and H Roy for questioning the role of *p*-Tyr editing. This work was supported by funding from the National Science Foundation (MCB 1052344 to MI; MCB 1052493 to BL), Ohio State University Center for RNA Biology Fellowships (to AR and NR) and NIH Training Grant Fellowships (T32 GM008512 and GM086252 to AM).

## Additional information

### Funding

| Funder | Grant reference number | Author |
|---|---|---|
| National Institutes of Health (NIH) | GM008512, GM086252 | Adil Moghal |
| National Science Foundation (NSF) | MCB1052344 | Michael Ibba |
| National Science Foundation (NSF) | MCB1052493 | Beth A Lazazzera |
| Ohio State University | Center for RNA Biology Fellowships | Andrei Rajkovic, Noah M Reynolds |

The funders had no role in study design, data collection and interpretation, or the decision to submit the work for publication.

## Author contributions
TJB, NMR, KFF, BL, MI, Conception and design, Acquisition of data, Analysis and interpretation of data, Drafting or revising the article; MR, AM, EM, AR, HK, FF, NH, Acquisition of data, Analysis and interpretation of data; CR, Conception and design, Acquisition of data, Analysis and interpretation of data

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
