## [Decision Letter]

Thank you for sending your work entitled 'Oxidation of cellular amino acid pools leads to cytotoxic mistranslation of the genetic code' for consideration at *eLife*. Your article has been favorably evaluated by a Senior editor, a Reviewing editor, and three reviewers.

The Reviewing editor has assembled the following comments to help you prepare a revised submission. All reviewers are enthusiastic about your study. One reviewer states: “Overall, the experiments were well described and the results clear. Most importantly, the conceptual advance is significant and likely to be of interest to the broad readers of *eLife*.” However, the reviewers feel two points need to be addressed before publication.

1) All reviewers felt the most interesting results in the manuscript focused on the *E. coli* story and found the yeast story distracting. Thus the authors should simply refer to the differences with yeast throughout, vaguely, and instead focus instead on the very tidy *E. coli* story so that the reader can take away this simple and elegant message.

2) The study would be strengthened if the authors could show that some m-Tyr indeed is incorporated to during oxidative stress since this is the main argument of the study. The reviewers understand that the low m-Tyr levels may present a challenge, but hope the experiment is do-able since data on m-Tyr incorporation into the proteome when cells are grown in m-Tyr are already provided.

---

## [Author Response]

We read your and the reviewers’ comments with interest and fully agree that the manuscript would greatly benefit from the revisions suggested. Consequently, a number of changes have been made which are detailed below, with reference to the specific comments of yourself and each of the referees.

*1) All reviewers felt the most interesting results in the manuscript focused on the* E. coli *story and found the yeast story distracting. Thus the authors should simply refer to the differences with yeast throughout, vaguely, and instead focus instead on the very tidy* E. coli *story so that the reader can take away this simple and elegant message*.

Yeast data on the UPR and on long-term survival have now been removed, and the only remaining results are those that refer to differences with *E. coli*.

*2) The study would be strengthened if the authors could show that some m-Tyr indeed is incorporated to during oxidative stress since this is the main argument of the study. The reviewers understand that the low m-Tyr levels may present a challenge, but hope the experiment is do-able since data on m-Tyr incorporation into the proteome when cells are grown in m-Tyr are already provided*.

Numerous attempts were made to address this question, but were unsuccessful; this is noted and explained in detail in the Results section.